# Modification of Breast Cancer Milieu with Chemotherapy plus Dendritic Cell Vaccine: An Approach to Select Best Therapeutic Strategies

**DOI:** 10.3390/biomedicines11020238

**Published:** 2023-01-17

**Authors:** Luis Mejías Sosa, Álvaro López-Janeiro, Alicia Córdoba Iturriagagoitia, Pablo Sala, Belén P. Solans, Laura Hato, Susana Inogés, Ascensión López-Díaz de Cerio, Francisco Guillén-Grima, Jaime Espinós, Susana De La Cruz, María Dolores Lozano, Miguel A Idoate, Marta Santisteban

**Affiliations:** 1Department of Pathology, Hospital Universitario Rey Juan Carlos, Gladiolo, 28933 Madrid, Spain; 2Department of Pathology, Clínica Universidad de Navarra, Pío XII 14 Avenue 36, 31008 Pamplona, Spain; 3Department of Pathology, Complejo Hospitalario de Navarra, Irunlarrea 3, 31008 Pamplona, Spain; 4Department of Medical Oncology, Breast Cancer Unit, Clínica Universidad de Navarra, Pío XII Avenue 36, 31008 Pamplona, Spain; 5Pharmacometrics and Systems Pharmacology, Department of Pharmacy and Pharmaceutical Technology, School of Pharmacy and Nutrition, Clínica Universidad de Navarra, Pío XII Avenue 36, 31008 Pamplona, Spain; 6Cell Therapy Unit, Department of Immunology and Immunotherapy, Clínica Universidad de Navarra, Pío XII Avenue 36, 31008 Pamplona, Spain; 7Navarra Institute for Health Research (IdisNA), Irunlarrea 3, 31008 Pamplona, Spain; 8Department of Preventive Medicine, Universidad de Navarra, Clínica Universidad de Navarra, Pío XII Avenue 36, 31008 Pamplona, Spain; 9Department of Medical Oncology, Complejo Hospitalario de Navarra, Irunlarrea 3, 31008 Pamplona, Spain

**Keywords:** breast cancer, dendritic cell vaccine, TILs, neoadjuvant chemotherapy, CD8 and triple negative

## Abstract

Background: The addition of dendritic cell vaccines (DCV) to NAC could induce immune responses in those patients with residual disease (RD) by transforming the tumor microenvironment. Methods: Core diagnostic biopsies and surgical specimens from 80 patients (38 in the vaccinated group plus NAC (VG) and 42 in the control group (CG, treated only with NAC) were selected. We quantify TILs (CD8, CD4 and CD45RO) using immunohistochemistry and the automated cellular imaging system (ACIS III) in paired samples. Results: A CD8 rise in TNBC samples was observed after NAC plus DCV, changing from 4.48% in the biopsy to 6.70% in the surgical specimen, not reaching statistically significant differences (*p* = 0.11). This enrichment was seen in up to 67% of TNBC patients in the experimental arm as compared with the CG (20%). An association between CD8 TILs before NAC (4% cut-off point) and pathological complete response in the VG was found in the univariate and multivariate analysis (OR = 1.41, IC95% 1.05–1.90; *p* = 0.02, and OR = 2.0, IC95% 1.05–3.9; *p* = 0.03, respectively). Conclusion: Our findings suggest that patients with TNBC could benefit from the stimulation of the antitumor immune system by using DCV together with NAC.

## 1. Introduction

Immune exclusion, immune ignorance or desert/cold breast cancer (BC) status is related to a worse outcome with standard chemotherapy as well as with immune checkpoint inhibitors; it is considered a predictive and prognostic factor. Thus, higher levels of tumor infiltrating lymphocytes (TILs) on the core diagnostic biopsy have been related to increased pathological complete responses (pCR) with neoadjuvant chemotherapy (NAC), and longer event-free survival (EFS) and overall survival (OS) in the more aggressive BC subtypes [1,2,3,4,5]. Additionally, the rise of TILs in RD after NAC has been linked to a better outcome, and it could be considered a surrogate marker for long-term treatment efficacy in triple negative breast cancer (TNBC) patients in the absence of pCR [5,6,7]. 

Therefore, the development of new immune strategies that increase TILs infiltration in tumors is needed. The study of biological changes in BC patients with RD after NAC opens a translational window in breast tumors with a poor prognosis. Immune checkpoint inhibitors have been evaluated in BC with outstanding results in combination with chemotherapy in the neoadjuvant scenario [8,9,10,11,12,13,14], with an increased toxicity profile but without information regarding biological impact on tumor milieu and systemic immunity. Dendritic cells, as the main directors of the immune system, have been studied for their role in the antigenic cross-presentation, a key step in T cytotoxic responses. Active cell therapy with dendritic vaccines (DCV) has shown tumor growth inhibition and T cell memory activation in preclinical models [15], as well as clinical improvement in BC patients without further toxicity in different scenarios [16,17]. Moreover, in the neoadjuvant scenario, immune checkpoint inhibitors have demonstrated clinical activity in both PD-L1 (positive and negative expression) populations, with a great advantage in PD-L1 positive tumors [18]. Although the benefit of DCV plus NAC has been shown regardless of PD-L1 expression, it has been detected more notably in PD-L1 negative BC population. So far, the implementation of DCV together with NAC in naïve BC patients could be of great benefit by increasing host systemic immunity, immunogenic tumoral cell death and immune infiltration in the tumor microenvironment [19,20,21].

The aim of this study is to evaluate if the addition of DCV to NAC could induce immune inflammation in BC patients with RD by transforming the tumor microenvironment, in order to identify which group of patients could benefit from this strategy and to select the best therapy in maintenance scenarios in upcoming studies.

## 2. Materials and Methods

### 2.1. Patients

Patients in the vaccinated group (VG) were recruited from 2011 to 2015 for the phase II non-randomized multicentric clinical trial NCT01431196, and for compassionate use of DCV. All these patients were diagnosed with non-HER2 overexpressing early BC and treated with the same NAC schedule, consisting of 4 cycles of dose-dense epirubicin plus cyclophosphamide with G-CSF support sequenced to 4 cycles of docetaxel every 21 days according to standard protocols, with the addition of DCV during taxane therapy. The control group (CG) was obtained from a historic cohort (2008–2015) treated at our center in the same way, but without vaccines. Demographic features were well balanced among both groups [22] (Table 1). No adjuvant chemotherapy was prescribed. Surgical management was performed after NAC and was followed by radiation therapy ± endocrine therapy if needed. In the adjuvant setting, patients in the VG also received intradermal DCV loaded with autologous tumor lysate, as described in our previous work [17,22].

### 2.2. Samples and Criteria for Analysis

Core-diagnostic biopsies and surgical specimens from 80 patients (38 in the VG and 42 in the CG) were selected. Pathologists did not know if the samples belonged to the VG or CG when they performed the quantification of TILs (blinded study).

Among the CG, two patients had multifocal/multicentric tumors in the breast. We were able to quantify TILs according to the characteristics of the core-diagnostic biopsies in the CG as follows: CD8 (37 samples), CD4 (39 samples) and CD45RO (36 samples). With respect to the surgical specimen, we studied 35 specimens for each marker. Regarding VG; 35, 34 and 34 core-diagnostic biopsies were available for CD8, CD4 and CD45ro biomarkers, respectively. Surgical specimens were 27, 28 and 26 for CD8, CD4 and CD45RO markers, respectively. Samples were classified by biological subtype according to the 14th St Gallen International Breast Cancer Conference [23]. Total pCR was considered as no infiltrating residual tumor (ypT0/Tis ypN0) in both the breast surgical specimen and lymph nodes according to American Joint Committee on Cancer, 8th Edition. Recommendations from the International Immuno-oncology Biomarker Working Group on BC were used in order to evaluate TILs in the stromal compartment related to tumoral cells. The denominator used to determine the percentage of stromal TILs was the area of stromal tissue occupied by lymphocytes, not the number of stromal cells [24]. Moreover, these recommendations were applied to immunohistochemistry to be able to measure CD8, CD4 and CD45RO lymphocytes in the stromal tumor area. Afterward, NAC markers were only measured on the samples with RD. Pathologists have not quantified TILs in the surgical specimens from patients that reached pCR, because it is not possible to accurately know if TILs were epithelial or stromal in this scenario. All the quantifications are in the stromal component; epithelial quantifications are not shown. No data for tertiary lymphoid structures were included in the study.

### 2.3. Immunohistochemistry

Immunohistochemistry was performed using formaldehyde-fixed and paraffin-embedded tissue sections from 3 to 4 mm thick. The tissue sections were stained by Autostainer Link 48 (Dako^®^) using a monoclonal Mouse Anti-Human CD8 (Dako^®^ Clone C8/144B), CD4 (Dako^®^ Clone 4B12) and CD45RO (Dako^®^ Clon UCHL1). Antigen retrieval was performed by PT link (Dako^®^) at a high pH at 98 degrees for 5 min.

### 2.4. Immunohistochemistry Measurement and Scoring

Immunohistochemistry was quantified using an automated cellular imaging system (ACIS III) in both the diagnostic and the surgical samples [25,26]. TILs quantification according to the international working group in hematoxylin and eosin samples was obtained by three pathologists (LM, AC, MI) from two different services with variability. These results are not shown in the manuscript.

The ACIS III system is an automated, bright-field microscope with a patented image processing and analysis software based on color detection and pattern recognition to evaluate cells or tissue sections stained with immunohistochemistry. Each slide was scanned at 10X. Stromal areas were evaluated according to the tumor burden within each sample. A minimum area of 0.23 mm^2^ and a maximum area of 3.10 mm^2^ by sample with an average area of 1.50 mm^2^ were analyzed. The score of each sample was obtained by dividing the number of brown pixels (stained by immunohistochemistry) by the total stromal area of the selected fields. Then, the result was transformed to a percentage value. Only areas with invasive breast carcinoma and without any processing defect were analyzed (Figure 1).

### 2.5. Vaccine Production

Fresh tumors were sent to the cell therapy laboratory. Tumor single-cell suspensions were obtained by mechanical disaggregation and then frozen and stored. Tumor lysate was obtained through four cycles of thawing and freezing, and then irradiated and stored at −20 °C. Seven days after dexamethasone termination, peripheral blood mononuclear cells were collected by leukapheresis. CD14+ cells were selected by immunomagnetic separation using a CliniMacs™ (Miltenyi Biotec, Bergisch Gladbach, Germany) following manufacturer’s instructions. These cells were cultured at 2 × 106 cells/mL in AIM-V (Gibco, Grand Island NY 14072,USA) supplemented with antibiotics, 1000 UI/mL of IL-4 (R&D Systems, Minneapolis, MN, USA) and 1000 UI/mL GM-CSF (Leukine, Genzyme Corporation, Bayer Healthcare, Seattle, WA, USA) in culture bags (Cellgenix, Gaithersburg, MD 20877, USA) at 37 °C in a humidified incubator. IL-4 (500 UI/mL) and GM-CSF (500 UI/mL) were further added to the medium on the 4th day, and cultured cells were harvested on the 7th day. These immature dendritic cells were adjusted at 107 cells/mL and pulsed with autologous tumor lysate (median 69.82 μg/mL, rank 27.9–75 μg/mL) for 2 h at 37 °C and 5% CO_2_. At that time, to induce dendritic cells maturation, 50 ng/mL of TNF-α (Beromun, Boehringer Ingelheim, Barcelona, España), 1000 UI/mL of IFN-α (Intron A, Schering Corporation, Kenilworth, NJ, USA) and 20 ng/mL Poli I:C (Amersham, GE Healthcare, Madrid, España) were added to the medium and cells were placed in culture bags at 2 × 106 cells/mL. Mature dendritic cells were harvested on the 8th day and frozen in aliquots following standard procedures until use. Briefly, the cells were resuspended in RPMI-1640 complete medium (500 mL RPMI-1640 (GIBCO, Life Technology, Madrid, España) + 50 mL of 10% FCS + 5 mL of l-Glutamine 200 mM (GIBCO, Life Technology, Madrid, España) + 5 mL Pen/Strep solution (solution with 10,000 U/mL Pen, 10 mg/mL Strep, GIBCO, Life Technology) at twice the desired cryopreservation concentration. The cryopreservation solution was prepared, containing 40% complete RPMI-1640, 40% FCS and 20% DMSO. The cryopreservation vials were placed in the cryopreservation box (5100 Crio 1° Freezing Container, Nalgene) and 500 microliters of the cell suspension was added to each vial; then, 500 μL of the cryopreservation solution was added and the final suspension was carefully mixed. The cryopreservation box was brought to a −80 °C freezer, and after 24 h, the cell vials were stored in a liquid nitrogen tank. Ten million cells was considered the optimal dose for each administration. The viability of cells was determined before and after freezing [22].

### 2.6. Statistical Analysis

Non-continuous data were compared by the Chi-square test and Fisher test. Normality was tested using Shapiro–Wilks Test. The Mann–Whitney U-test and the Wilcoxon test were employed to study unpaired non-parametric variables and paired non-parametric variables, respectively. Sensitivity, specificity, predictive values, logistic regression and receiver–operator characteristic (ROC) curves were calculated. Data were analyzed with the SPSS statistical software (version 20.0 for Windows), OpenEpi: Open Source Epidemiologic Statistics for Public Health, Version 3.01 and R software Version 4.0.1. *p*-values ≤ 0.05 were considered significant, and 95% confidence intervals were calculated.

## 3. Results

### 3.1. Triple Negative Vaccinated Breast Cancer Showed the Highest Pathologic Complete Response Compared with the Groups Corresponding to Luminal A and Luminal B Tumors

The pathologic complete response rate was superior among vaccinated patients (28.9% versus 9.1%, *p*  =  0.03, absolute increment of 19%). According to subtype, triple negative (TN) BC patients experienced the highest pCR (50% for VG versus 30.7% for CG, *p*  =  0.25, absolute benefit of 19%), with modest responses for luminal B types (16.6% for vaccinated versus none in CG, *p*  =  0.15). No pCR was seen in luminal A tumors from any group.

### 3.2. Triple Negative Vaccinated Breast Cancer Showed the Highest Rate of CD8 TILs before and after Chemotherapy

All the patients in this study with a paired-sample analysis of TILs are those with RD in the surgical specimen (those who reached pCR are excluded due to the impossibility of quantifying lymphocytes related to tumor cells). The biologic subtype with the highest rate of CD8 TILs in paired samples in both groups was the TNBC (*p* < 0.0001). When data were analyzed according to biologic subtype, a trend toward a CD8 TILs rise in TNBC samples was observed after NAC plus DCV, changing from 4.48% (0.48–16.26) in the diagnostic biopsy to 6.70% (0.76–11.66) in the surgical specimen, not reaching statistically significant differences (*p* = 0.11). On the contrary, TNBC patients in the CG showed a TILs drop from 2.71% in the biopsy to 0.18% in the surgical specimen (*p* = 0.5), see Table 2. We also found that 66.7% (4/6) of TNBC patients from VG registered a rise in TILs after treatment compared with 20% (1/5) of TNBC patients in the CG (*p* = 0.24). This phenomenon is not observed in the other biologic subtypes and is clarified in (Figure 2). On the other hand, when we analyze both groups without specifying the biologic subtype, CD8 TILs decreased after NAC ± DCV in tumors compared with initial values. The median CD8 TILs was 1.43% (0.03–13.29%) before NAC compared with 0.58% (0.03–39.28%) after NAC (*p* = 0.51) in the CG; and 1.33% (0.19–16.26%) versus 0.59% (0.14–11.66%) (*p* = 0.42) in the VG. We found no significant findings in the quantification of CD4 and CD45 TILs (data not shown).

### 3.3. CD8 TILs Cut-Off Point in the Core-Diagnostic Biopsy Could Be Used for Selecting Patients for Breast Cancer Vaccination

An association between CD8 TILs before NAC and pCR in VG was found in both the univariate and multivariate analysis (OR = 1.41, IC95% 1.05–1.90; *p* = 0.02, and OR = 2.0, IC95% 1.05–3.9; *p* = 0.03, respectively). However, this association disappeared in the CG in the univariate OR = 1.18, (IC95% 0.90–1.55; *p* = 0.22); and in the multivariate analysis, OR = 1.11 (IC95% 0.68–1.81; *p* = 0.65)). No association between CD4 or CD45RO T cells before NAC and pCR in both analyses was observed in the CG and VG (data not shown). The association between TILs and pCR disappears when the samples are separated by biologic subtype, regardless of therapeutic arm.

In relation to pCR, the ROC curve establishes a cut-off point of 4% for CD8 TILs in the diagnostic samples (Figure 3). With this cut-off point, a 92.6% sensitivity (IC95% 76.63–97.94), 75% specificity (IC95% 40.93–92.85), 92.6% positive predictive value (PPV) (IC95% 76.63–97.94) and a 75% negative predicted value (NPV) (IC95% 40.93–92.85) were obtained in the VG. Using the same cut-off point in the CG, a similar sensitivity of 87.9% (IC95%:72.67–95.18) and a PPV of 93.55% (IC95%: 79.28–98.21) was reached, although the specificity and NPV were 50% (IC95%: 15–85) and 33.33% (IC95%: 9.6–70), respectively. Differences between both groups were not statistically significant (*p* = 0.20).

Regarding outcome, we have calculated the percentage of patients that relapse regarding the amount of TILs at diagnosis with a median follow-up of 8 years. With the established 4% TILs cut-off in the diagnostic specimen, 12.5% (1/8) patients with ≥4% CD8 TILs in the VG and 16.6% (1/6) in the CG have disease progression. Among those patients with <4% TILs, 7.4% (2/27) patients in the VG and 19% (8/42) in the CG relapsed. There is a trend toward decreasing relapse (Fisher test, *p* = 0.30) within the VG. No comparable results were found with CD4 and CD45RO TILs.

### 3.4. No Correlation between Quantified TILs and Pathological Response Was Observed in the Breast Cancer Surgical Samples after Vaccination

We also correlate quantification of TILs in residual samples with the level of pathological response categorized by Miller and Payne (3–4 M and P grades). Although sample size was limited, we have not seen correlation among both variables (Spearman’s rank correlation *p* = 0.77). A total of 23 vaccinated patients with residual disease reached moderate to high pathological responses (3–4 M and P grades) compared with four patients with poorer pathological responses (1–2 M and P grades).

## 4. Discussion

To our knowledge, this is the first report that analyzes changes in tumor milieu in early non-HER2 overexpressing BC patients treated with NAC ± immunotherapy and identifies good responders to the experimental therapy based on CD8 expression in the needle core biopsy, with an established 4% cut-off point. Moreover, we have analyzed biomarkers other than CD8 in paired samples without a clinical value. Although non-significant, we have also appreciated a trend toward a better outcome when TILs were higher in RD, with fewer relapse events in the VG. As expected, higher TILs in the diagnostic biopsy translate to higher pCR rates. Loibl et al. demonstrated that stromal TILs were significant for pCR in both the control and the durvalumab cohorts in an exploratory biomarker analysis, although they did not show the same results with intratumoral TILs. However, they also performed paired analysis in core biopsies as well as in the postwindow biopsy, and they could conclude that those patients that have an increase in TILs in the durvalumab arm (but not in the control arm) predict pCR [12].

The clinical impact of these findings could be relevant to improving the selection of (1) patients that could benefit from the addition of DCV to NAC; and (2) an optimal maintenance therapy in TNBC subtypes with RD, based on the expression of CD8 and CD4 markers. New designs of clinical trials with different options among standard capecitabine [8,27] versus immunotherapy based on immune checkpoint inhibitors [18] or on DCV are needed. In the Keynote 522 study, it appears that patients with residual cancer burden 2 are the ones who receive the highest benefit from pembrolizumab maintenance [28].

Modifications on the tumor niche that reflect immune activation after DCV help it to be open wide to less toxic and more specific immunotherapy.

Primary TNBC patients reach higher levels of TILs than the luminal subtypes in our study, as described before [5]. This could be in part due to specific mutational signatures, copy number variations, stromal metagenes and clonal heterogeneity of the TNBC subtype [29,30,31].

Furthermore, we have shown a non-significant increase in stromal TILs after combined therapies with NAC plus DCV in up to 66% of TNBC patients, but not in the luminal subtypes. Increased TILs in RD could illustrate a reinforcement of the immune niche produced by DCV in BC patients. It can be suggested that vaccines could produce an increase in CD8 T cells post-treatment in TNBC. Waks et al. found that luminal BC are less enriched in CD8 TILs than other BC subtypes, and this cell population decreased after NAC [32].

Nowadays, there is not enough knowledge about how immunotherapy could change the BC microenvironment. Results of the studies that have worked with paired samples (before and after conventional NAC and no immunotherapy) remain controversial, suggesting different roles of immune cell populations in carcinogenesis, response to therapies, tumor progression and crosstalk among the tumor and the microenvironment. While some reports described a reduction of CD8 TILs in cancer milieu after NAC [32,33,34,35], others showed an increase on TILs count [6,36,37] or an inversion on the CD4/CD8 ratio [4]. Luen et al. described a rise in TILs level in 48% of the patients and a decrease in 47% of the patients in RD after NAC [38].

The input of immunotherapy in BC patients in the neoadjuvant scenario looks for an improvement within the tumor (higher pCR), the niche (hottest tumors) and the systemic immune surveillance [21], as well as with EFS [18,39]. Dendritic cell-based adjuvant immunotherapy has already shown a gain in CD8 T cells in peripheral blood in non-luminal BC, with an encouraging progression-free survival improvement suggesting benefits of systemic immunity [16]. Vaccination is more effective in the prevention of tumor growth, and most probably, the clinical advantage could be more relevant in patients with a small tumor burden and with a preserved immune system (naïve of therapies) than in large tumor burden patients, including metastatic scenarios and those with an exhausted immune system. Thus, the immune cell profile in the niche of primary versus metastases of BC patients is different and more active in the former group [40,41].

Recent efforts have been made to standardize the quantification of TILs and to produce reliable results [42,43,44,45]. Some authors pointed out that the reproducibility of TILs evaluation improves when the categories are simplified to low versus high TILs based on the concept of lymphocyte-predominant breast cancer phenotype [46], but the standard cut-off points need to be defined. The incorporation of digital analysis (ACIS III) provides a better quantification of the stromal TILs, avoiding interobserver bias. Additionally, another aspect to be considered is the selection of the area to be evaluated and how many fields should be selected. Recent works support the fact that the average lymphocyte score from a single biopsy of a tumor is reasonably representative of the whole cancer [47]. Our results showed that a cut-off point of 4% in CD8 TILs in the diagnostic specimen could predict a better pathologic response when DCV is added to conventional NAC, with higher responses and a trend toward a lower rate of relapse (not significant). As it can be observed in Figure 1, this cut-off could be considered a relatively high density of CD8 cells in the stromal tumor. This optical microscope immunohistochemical image could be used as a reference for what we consider to be the minimal density of cells that correlates with response to vaccines.

Qingzhu et al., found in their meta-analysis that memory T lymphocyte infiltration of a tumor site could serve as an indicator for progression-free survival and OS prognosis in patients with malignant tumors [48]. Additionally, Yajima et al., found a relationship between high CD45RO (i.e., memory T lymphocyte and marker of activated T cells) expression and a lower pathological stage in BC patients [49]. The importance of CD4 TILs is not clear in the BC arena. No association between CD4 and pCR was found in our study, although in the CG, CD4 TILs significantly decreased in patients after NAC (*p* = 0.04) compared with those who received the experimental therapy and remained stable (*p* = 0.24) (Table 2). However, most of the studies correlate CD4 or the CD4/CD8 ratio with an increased pCR and better survival [5,34]; this fact strengthens the role of DCV.

We have not looked for tertiary lymphoid structures in this study, although they are seen occasionally in the samples. Their neogenesis is not specific to tumors and immune therapies applied in oncology, so they also appear in autoimmune and infectious diseases, transplanted solid organs and inflammatory disorders, as well as after preventive vaccines (e.g., HPV).

Although more studies are needed to establish the benefit of DCV addition to chemotherapy, the identification of predictive and prognostic biomarkers to select patients that could benefit from the addition of immunotherapy to the standard systemic chemotherapy is key to develop more efficient therapeutic strategies [50]. The relationship between lymphocyte-predominant breast cancer patients in the case of the TNBC [51] and the pCR, progression-free survival and OS is clearly established [52,53,54]. Nonetheless, an outstanding selection of good responders to immunotherapy is complex [55], and the study of biomarkers in RD after NAC plus immunotherapy as well as in peripheral blood is mandatory to improve this limitation. In our previous work, we found phenotypic changes in the peripheral blood of 18 patients in the VG. Our results showed a significant decrease in myeloid-derived suppressor cells (MDSC) (16.5% versus 7.9%, *p*  <  0.01) and an increase in the NK cells (8% versus 14%, *p*  <  0.01) after treatment. Moreover, we found an increase of the activation marker HLA-DR in CD4 (3.8% versus 8.3%, *p*  =  0.002) and in CD8 T cells (10.7% versus 18.4%, *p*  <  0.01), and a significant decrease in the expression of PD-1 (13.8% versus 9.7%, *p*  <  0.05) and TIM-3 (4.7% versus 3.3%, *p*  <  0.05) in CD8 T cells after DCV [21]. Regarding the EFS and OS, the median follow-up was 6.96 years in the VG and 9 years in the CG. No significant differences were observed for event EFS or for OS. The percentage of patients at 5 years who were alive with disease progression was 12.82% in the VG versus 14.35% in the CG, whereas after 7 years, it was 17.08% in the CG and 19.70% in the VG. At 5- and 7-year follow-up, patients alive in the CG were 90.41% and 87.58%, respectively, and 94.87% and 91.36% in the VG, respectively [21].

Regarding functional assays, an increase in peripheral blood mononuclear cells’ proliferation and IFN-γ production with specific tumor lysate was detected in DCV-treated samples compared with basal ones in 69% (*p*  =  0.03) and 74% (*p*  =  0.15) of the patients, respectively. However, no correlation was found between the clinical response and the increase in proliferation or in IFN-γ production (data not shown). Changes in cytokines profiles were evaluated in the serum samples of 20 patients after DCV, highlighting a decrease in IL-6 levels (1.9 versus 1.4 pg/mL, *p*  <  0.05) [21].

Evaluation of tumor responses to immune strategies using imaging techniques has become tricky because changes in tumor burden need new response criteria based on special guidelines (iRECIST) [56]. In this way, biologic markers in the blood, the tumor and its milieu could contribute to more specific information than imaging markers regarding patient selection for immune strategies in the early BC arena.

## 5. Conclusions

On biopsy, a possible 4% cut-off point of CD8 TILs in a TNBC subtype could help to establish which patients can benefit from DCV added to NAC. A trend toward a lower rate of relapse in diagnostic biopsies enriched with TILs has been also shown. Our findings suggest that patients with TNBC especially benefit from the stimulation of the antitumor immune system by using DCV pulsed with tumor antigens with an excellent tolerance. Deeper studies based on immune-profiling genomic panels in the tumor and immune cell populations on peripheral blood are still ongoing within our patients, and could help us to elucidate the biological behavior of BC as well as the benefits of adding immunotherapy to conventional NAC. Combined immune approaches potentiating the immune system, with DCV added to the blockade of immune checkpoints together with NAC, should be tested in clinical trials in this selected population.

## Figures and Tables

**Figure 1 biomedicines-11-00238-f001:**
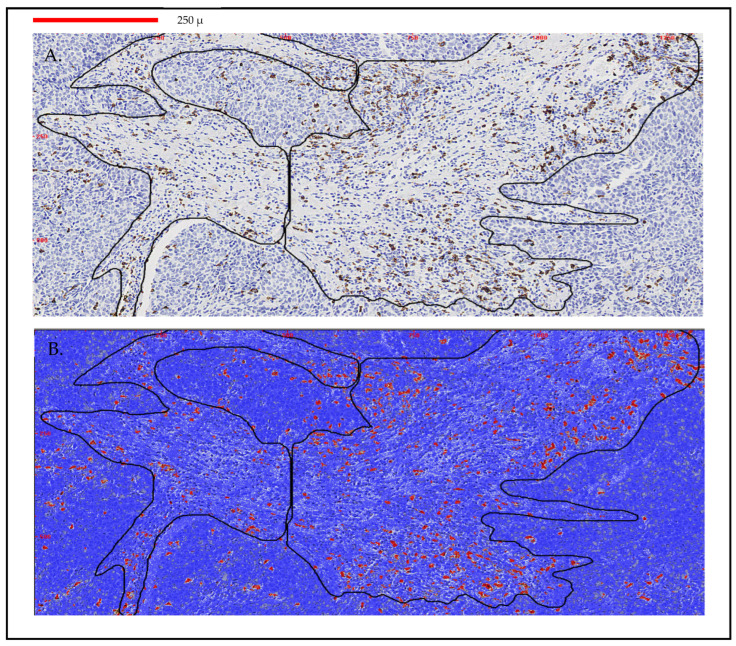
Immunohistochemistry measurement and scoring with ACIS III in a breast cancer sample. (**A**) Image analysis by ACIS III of a biopsy with two stromal areas selected (0.43 mm^2^). (**B**) TILs quantification by ACIS III in red CD8 lymphocytes 5%. Scanned at 10×; magnification of 50%. Line identifies 250 μm.

**Figure 2 biomedicines-11-00238-f002:**
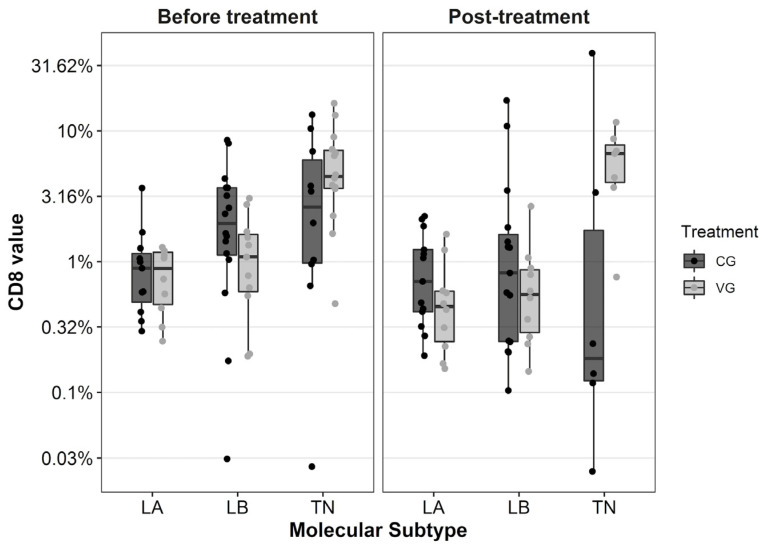
Boxplot diagram with the distribution of CD8 TILs before and after neoadjuvant chemotherapy according to biological subtype and group of treatment. Each dot represents the value for each patient. The Y-axis is shown in logarithmic scale. TILs: tumor infiltrating lymphocytes, LA: luminal A, LB: luminal B, TN: triple negative BC, CG: control group, VG: vaccinated group.

**Figure 3 biomedicines-11-00238-f003:**
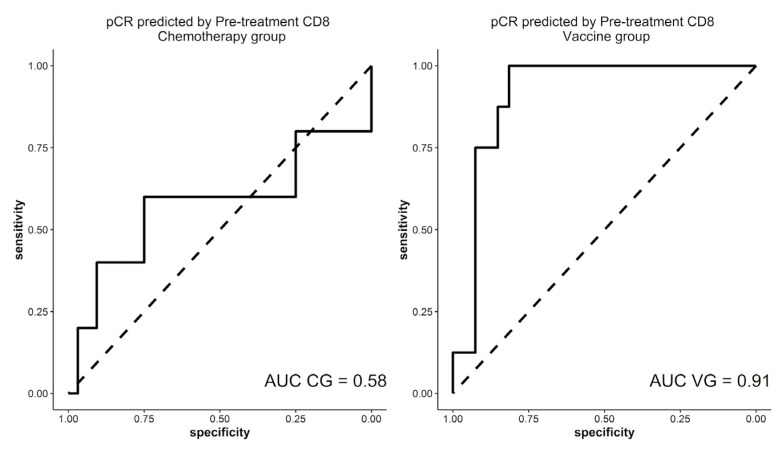
ROC curves in both therapeutic groups that correlate CD8 TILs in the diagnostic specimen and pCR. TILs: tumor infiltrating lymphocytes, pCR: pathological complete response, AUC: area under the curve, CG: control group, VG: vaccinated group.

**Table 1 biomedicines-11-00238-t001:** Patient characteristics corresponding to control and vaccinated groups.

Characteristics	Control Group (CG) N 42	Vaccinated Group (VG) N 38
Age ^a^ (range)	55.31 (26–84.35)	45.68 (36.15–74.48)
Menopause ^b^ (%) Pre-menopause	18 (42.9)	28 (71.8)
Post-menopause	24 (57.1)	11 (28.2)
Lymph node status * (%) Negative	24 (53.3)	19 (50)
Positive	21 (46.7)	19 (50)
Biological subtype * (%) Luminal A	14 (31.1)	10 (26.3)
Luminal B	18 (40)	12 (31.6)
Triple negative	13 (28.9)	16 (42.)
Pathological CR *(%) Yes	4 (8.9)	10 (26.3)
No	41 (91.1)	28 (73.7)
Treatment schedule (%) ddEC → D CBDCA added to D	39 (92.85) 3 (7.15)	35 (92.10) 3 (7.90)
Total dose (mgr) E (mean, range) D (mean, range)	380.56 (298–406) 340.78 (282–398)	382.46 (352–409) 333.54 (274–400)
Radiotherapy Yes No	41 (97.61) 1 (2.39)	36 (94.73) 2 (5.27)

^a^ Median values. ^b^ According to the number of patients. * According to the number of tumor samples. pCR pathological complete response. N: number; dd: dose dense; E: epirubicin; C: cyclophosphamide; D: docetaxel; CBDCA: carboplatin.

**Table 2 biomedicines-11-00238-t002:** Quantification of CD8 TILs according to biologic subtypes in tumor samples.

TILs	PreNAC % (Range)	PostNAC (Range)	*p*
CD8 in CG	1.43 (0.03–13.29)	0.58 (0.03–39.28)	0.51
LA	0.89 (0.29–3.66)	0.59 (0.19–2.23)	0.79
LB	1.98 (0.03–8.49)	0.81 (0.10–17.09)	0.69
TN	2,71 (0.03–13.29)	0.18 (0.03–39.28)	0.50
CD8 in VG	1.33 (0.19–16.26)	0.59 (0.14–11.66)	0.42
LA	0.90 (0.25–1.29)	0.45 (0.15–1.62)	0.13
LB	1.08 (0.19–3.06)	0.56 (0.14–2.65)	0.07
TN	4.48 (0.48–16.26)	6.70 (0.76–11.66)	0.11

Results are expressed as median values with range. CG: Control Group; VG: vaccinated group; LA: luminal A, LB: luminal B, TN: Triple negative.

## Data Availability

The datasets generated and analyzed during the current study are available from the corresponding author on reasonable request.

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
