# Peer review of "Modification of Breast Cancer Milieu with Chemotherapy plus Dendritic Cell Vaccine: An Approach to Select Best Therapeutic Strategies"

_biomedicines, 2023, doi:10.3390/biomedicines11020238_

Round 1
Reviewer 1 Report
In present studies, authors were elucidated the evidences that dendritic cell-based vaccination with common neoadjuvant chemotherapy were more effective than traditional NAC single therapy. To investigate the statistic correlation, authors were selected two group (NAC only and NAC + DCV) and tried qualification of T-cell class markers and compared luminal A and B from biopsied tumor tissue by immunohistochemistry.
Author Response
Response to Reviewer 1 Comments
Thank you so much for your opinion. We have included a more detailed explanation of vaccine production in the methodology section. In addition, we have indicated the purpose or scope that concerns each result. Regarding the conclusion, we have tried to rewrite it in a more concrete way.
Reviewer 2 Report
The article titled "MODIFICATION OF BREAST CANCER MILIEU WITH CHEMOTHERAPY PLUS DENDRITIC CELL VACCINE: AN APPROACH TO SELECT BEST THERAPEUTIC STRATEGIES", authored by Luis Mejías Sosa ana colleagues, provides evidence that connect tumor microenvironment with therapeutic opportunities of breast vancer. The report is interesting and with novel evidence to support the autors' conclusions. Comments and Suggestions for Authors are following: 1) the manuscript is well-structured. However I would suggest that the methods should be described more extensively. The same is true in the results section, where is necessary to add a purpose before every result presentetion and a clear conclusion in the end. 2) I would discourage the use of such a lerge number of abbreviations in the main text. I suggest to ommit abbreviations that are used only a couple of times and a list of abbreviations in the end of the main text.Author Response
Response to Reviewer 2 Comments
Thank you very much for your comments and suggestions. We are deeply grateful. Here we have our answers:
Point 1. The manuscript is well-structured. However I would suggest that the methods should be described more extensively. The same is true in the results section, where it is necessary to add a purpose before every result presentation and a clear conclusion in the end.
We have added an extensive explanation on the production of vaccines, based on the work of Dr. Inogés and we also added the purpose before every result presentation with a clear conclusion.
Reference:
Inogés S, Tejada S, de Cerio AL, Gállego Pérez-Larraya J, Espinós J, Idoate MA, Domínguez PD, de Eulate RG, Aristu J, Bendandi M, Pastor F, Alonso M, Andreu E, Cardoso FP, Valle RD. A phase II trial of autologous dendritic cell vaccination and radiochemotherapy following fluorescence-guided surgery in newly diagnosed glioblastoma patients. J Transl Med. 2017 May 12;15(1):104. doi: 10.1186/s12967-017-1202-z. PMID: 28499389; PMCID: PMC5427614.
Point 2. I would discourage the use of such a large number of abbreviations in the main text. I suggest omitting abbreviations that are used only a couple of times and a list of abbreviations at the end of the main text.
We have revised the abbreviations, and we have reduced them. In addition, we have added a list of abbreviations.
Round 2
Reviewer 2 Report
The authors have presented an improved version of their manuscript that answers most of my comments.
However, the results section must be improved to include a clear purpose and conclusion for each result. This is missing in the presented revision.
Author Response
Point 1. The results section must be improved to include a clear purpose and conclusion for each result. This is missing in the presented revision.
Dear reviewer,
We have modified the titles corresponding to results, clearly establishing what to expect from each section. We would also like to comment that we have modified paragraph 3.1 of results. In this paragraph we cited the results of OS and EFS from our previous work, we believe it is more appropriate to comment on these findings in the discussion section.